# Efficacy of Management and Monitoring Methods to Prevent Post-Harvest Losses Caused by Rodents

**DOI:** 10.3390/ani10091612

**Published:** 2020-09-09

**Authors:** Inge M. Krijger, Gerrit Gort, Steven R. Belmain, Peter W. G. Groot Koerkamp, Rokeya B. Shafali, Bastiaan G. Meerburg

**Affiliations:** 1Livestock Research, Wageningen University & Research, 6700 AH Wageningen, The Netherlands; bmeerburg@kad.nl; 2Farm Technology Group, Wageningen University & Research, P.O. Box 16, 6700 AA Wageningen, The Netherlands; peter.grootkoerkamp@wur.nl; 3Dutch Pest and Wildlife Expertise Centre (KAD), Nudepark 145, 6702 DZ Wageningen, The Netherlands; 4Biometris, Wageningen University & Research, 6700 AA Wageningen, The Netherlands; gerrit.gort@wur.nl; 5Natural Resources Institute, University of Greenwich, Central Avenue, Chatham Maritime, Kent ME4 4TB, UK; s.r.belmain@greenwich.ac.uk; 6Association for Integrated Development-Comilla (AID-COMILLA), Raghupur, Rajapara, Jagannathpur, Comilla Sadar, Comilla 3500, Bangladesh; aidshafali@yahoo.com

**Keywords:** grain store, rodent management, post-harvest losses, *Bandicota bengalensis*, *Rattus rattus*, rice

## Abstract

**Simple Summary:**

The presence of pest rodents around food production and storage sites is one of many underlying problems contributing to food contamination and loss, particularly influencing food and nutrition security in low-income countries. By reducing harvest losses by rodents, millions of food-insecure people would benefit. As there are limited data on post-harvest rice losses due to rodents, our objectives were to assess stored rice losses in local households and two rice milling factories in Bangladesh. We also wanted to monitor the effect of different rodent control strategies on stored rice losses over a period of two years (2016 and 2017). Four control strategies were tested, (i) untreated control, (ii) use of domestic cats, (iii) use of rodenticides, (iv) use of snap-traps. In total, 210 rodents were captured from inside people’s homes, with *Rattus rattus* trapped most often, followed by *Mus musculus* and *Bandicota bengalensis*. In the milling stations, 68 rodents were trapped, of which 21 were *M. musculus*, 19 *R. rattus*, 17 *B. bengalensis*, 8 *Rattus exulans*, and 3 *Mus terricolor*. In 2016, losses from standardised baskets of rice within households were between 13.6% and 16.7%. In 2017, the losses were lower, ranging from 0.6% to 2.2%. Daily rodent removal by trapping proved to be most effective to diminish stored produce loss. The effectiveness of domestic cats was limited.

**Abstract:**

The presence of pest rodents around food production and storage sites is one of many underlying problems contributing to food contamination and loss, particularly influencing food and nutrition security in low-income countries. By reducing both pre- and post-harvest losses by rodents, millions of food-insecure people would benefit. As there are limited quantitative data on post-harvest rice losses due to rodents, our objectives were to assess stored rice losses in local households from eight rural communities and two rice milling factories in Bangladesh and to monitor the effect of different rodent control strategies to limit potential losses. Four treatments were applied in 2016 and 2017, (i) untreated control, (ii) use of domestic cats, (iii) use of rodenticides, (iv) use of snap-traps. In total, over a two-year period, 210 rodents were captured from inside people’s homes, with *Rattus rattus* trapped most often (*n* = 91), followed by *Mus musculus* (*n* = 75) and *Bandicota bengalensis* (*n* = 26). In the milling stations, 68 rodents were trapped, of which 21 were *M. musculus*, 19 *R. rattus*, 17 *B. bengalensis*, 8 *Rattus exulans*, and 3 *Mus terricolor*. In 2016, losses from standardised baskets of rice within households were between 13.6% and 16.7%. In 2017, the losses were lower, ranging from 0.6% to 2.2%. Daily rodent removal by trapping proved to be most effective to diminish stored produce loss. The effectiveness of domestic cats was limited.

## 1. Introduction

The fight against hunger persists, with the number of undernourished people continuing to rise. In 2017 about 820 million people were undernourished globally [1]. The Food and Agriculture Organisation (FAO) defines undernourishment as the daily energy intake of a person being too low to meet their daily minimum dietary energy requirements (kcal/day/person). Southern Asia has the highest undernourishment rate, with an estimated 275 million people suffering from hunger [1]. In Bangladesh the proportion of undernourished people in 2017 was around 15% of the total population, which is almost 25 million people [1]. Asia produces more than 90% of the global rice production, with rice accounting for approximately 60% of the daily caloric intake, on average, across Asia [2]. One contributing factor to food insecurity is the presence of rodents. On a yearly basis rodents cause 5–10% loss to rice production in Asia, which leads to a worldwide estimated loss of 11 kg of food per person per year [2]. By sustainably reducing pre- and post-harvest losses by rodents, nearly 280 million undernourished people could meet their daily energy requirements [3].

In 2018, Bangladesh produced 53.6 million tons of paddy rice [4], where a 10% post-harvest rice loss due to rodents equated to an annual loss of 5.36 million tons. Singleton [2] and Meerburg et al. [5] state that reports of up to 15–20% post-harvest grain losses due to rodents are not uncommon. Unfortunately, there is little quantitative data on Asian post-harvest losses to cereals due to rodents [2,6,7,8,9]. From previous studies it is known that rodents do play a significant role in post-harvest losses in Asia, but only a few recent publications [8,9,10] have provided information on the magnitude of the post-harvest rice losses within villages in Southern Asia. In Myanmar, Htwe et al. [9] calculated that the total amount of grain that was lost due to rodents was enough rice to feed local households for 1.6–4 months. Belmain et al. [8] showed that farmers without rodent management on average lose 2.5% of their stored rice stocks, but when applying rodent management they reduce the loss to 0.5%. Therefore, the first objective of the current study is to assess how large post-harvest losses in Bangladesh are in local households and in rice milling stations.

As rodent management can reduce stored product losses [11], the need to implement or improve cost-beneficial rodent management strategies based on sound ecological research to understand their patterns of behaviour and feeding becomes essential [12,13]. For example, measuring the actual impact of rodents within stored rice facilities is difficult as these animals not only eat rice [14], but also hoard and move rice to other locations [15] and damage and contaminate the grain [8]. As rodents forage from several different habitats, their feeding from any given grain store cannot be easily estimated through simple estimations of rodent population density [16,17]. Although pest rodent presence is considered a problem across many rural farming communities (e.g., by damaging clothes, blankets, transmitting disease, eating and contaminating stored rice), rodent management is usually applied too late [18,19]. Rodent control is mostly practised once damage to crops or stored produce becomes visible [12,20], whilst rodent control in rural Asian environments relies mainly on the use of rodenticides [6,21]. Other management methods which can be applied are trapping, habitat management (e.g., proofing, sanitation), and biocontrol (e.g., wild or domestic predators, rodent pathogens) [18,22]. As a second objective of the study, we wanted to assess the efficacy of different rodent management methods capable of reducing post-harvest losses under local contexts.

## 2. Materials and Methods

### 2.1. Study Locations

Research was conducted over 2016 and 2017 in the South-East region of Bangladesh. The South-East region is one of the most important rice growing areas of Bangladesh and was selected because the area represents typical rice production practices and enabled us to choose villages that were relatively close to each other to ensure similar abiotic/biotic parameters that would reduce potential bias in environmental parameters across villages. Choices were further informed by local collaborators enabling access and permissions for ethical clearance. In total, two rice milling stations (Modern Rice Milling Unit, and Sonali Rice mill, respectively 23°27’26.2” N 91°10’19.3” E and 23°20’55.7″ N 91°15’38.3″ E) and eight villages participated in the study. The selected villages were: (A) Lakhshmipur, (B) Comalla, (C) Kadamtoli, (D) Monahpur, (E) Maruali, (F) West-Maruali, (G) Nagarkandi, and (H) Baro Char, and were together with the rice mills situated in the Chittagong division, Comilla district, Comilla sadar upazila (all villages are located within 10 km of 23°27’23.0″ N 91°10’20.6″ E, Figure 1).

Bangladesh has a subtropical monsoon climate, which is characterised by broad annual variations in rainfall, temperatures, and humidity [23,24]. The selected sites all experience the same climate and monsoon rainfall cycles between June and October. In Bangladesh there are two to three crops per year (depending on the climate and irrigation possibilities during the dry period), with most farmers planting rice [25]. The size of the communities selected was between 75 and 150 households per village. Ethical approval was obtained through project partner AID-Comilla, which explained activities in the local language, gaining consent from each household involved. The owners of the rice mills were part of the project team and agreed to use their properties and buildings for this study.

For every village, ten households were randomly selected, with the pre-condition that the household stored paddy rice for at least three months after each harvest and consented to participate in the study. The most commonly used storage methods in Bangladesh for paddy (~80% of households) are jute sacks or baskets made from woven reeds, bamboo, and/or wood, which are usually left uncovered and positioned in bedrooms or other living areas. Other storage methods are plastic barrels or steel drums (often without a lid), which are more expensive and are present in ~20% of households [26]. Storage trials took place during both the wet (June, July, August) and dry seasons (October, November, December), with one replication (2016 and 2017, Figure 2). Assignment of treatments to villages was done randomly.

In order to reduce the possibility of trial activities influencing rodent populations, data collection in the wet season took place in four different villages (A–D) from those involved during the dry season (E–H).

### 2.2. Assessment of Stored Rice Losses

To assess the rice losses due to rodents, we used the method developed by Belmain et al. [20] as the basis for both the rice milling stations and households. Baskets made from woven reeds and bamboo were purchased in a local market. The baskets were 20 cm deep, had a base diameter of 21 cm, and a diameter of 41 cm at the open top (Figure 3). Each basket was filled with 5 kg of threshed paddy rice, and each selected household received one rice basket to determine the loss due to rodents (*n* = 10 per village) for a period of three consecutive months. Baskets were placed in the same area as the family food store, and every fortnight the baskets were weighed and moisture content was measured using a portable grain moisture meter (Model GMK 303RS; G-WON HITECH Co. LTD, Seoul, Korea). In the two rice milling stations, 10 rice baskets filled with threshed rice were randomly placed in the paddy rice storage warehouse for a period of nine months (July 2016–March 2017). The rice baskets in the mills were also weighed and moisture content was measured every fortnight. In contrast to the procedure described in Belmain et al. [20], baskets were not restocked after weight measurements to avoid potential problems in variable grain quality in the baskets compared to rice in farmer granaries that could affect rodent feeding behaviour.

Weight losses of the rice in the baskets could be influenced by potential moisture changes. To correct for those changes in moisture content, the following formula was used:(1)$$Wa=Wi*\left(100-MCi\right)/(100-MCf)$$
with *Wa* being the adjusted weight, *Wi* initial weight, *MCi* = initial moisture content (%), and *MCf* = the final moisture content (%). All results are reported as adjusted weights.

### 2.3. Monitoring Rodent Presence

Rodent presence was monitored before, during, and after the treatments in both the rice milling stations and in the villages for two consecutive days each fortnight using Giving up Densities (GUD) and tracking tiles. To measure GUDs, open plastic trays of 30 × 20 × 8 cm were filled with approximately 4 cm local sand within which 25 peanuts were randomly buried [27,28]. The sand was sieved the next morning in order to count the peanuts eaten, and all trays were restocked to repeat the procedure over two consecutive nights every fortnight. Each household received one tray, which was placed in an area near obvious signs of rodent presence (faeces, holes, damage to storage structures). Tracking tiles (Figure 4) were used to passively monitor rodent activity and consisted of white ceramic wall tiles (20 × 30 cm) that were blackened with soot using a smoking paraffin lamp. Two blackened tiles were placed in each household for two consecutive days each. The percentage area marked by rodent footprints was determined by placing a transparent plastic sheet marked into 16 cells on top of the tile (Figure 4B). The number of cells with rodent footprints was expressed as a percentage of the total number of cells. By calculating the percentage of the tile covered with footprints the relative amount of rodent activity could be measured [29]. After each count, tiles were re-blackened.

### 2.4. Rodent Control Measures in Villages

Ethical approval and permission for the work were secured through the owners of the mills and the individuals involved in all the communities. All staff followed international guidelines on the handling of wild mammals in field research [30] and according to the Netherlands code of scientific practice. Although the animals used were not laboratory animals, the NCad opinion on “alternative methods for killing laboratory animals” was followed, as provided by the Netherlands National Committee for the protection of animals used for scientific purposes [31]. There were four treatments assessed: (I) control (no treatment); (II) place 20 domestic cats per village. Cats were maintained two each at the 10 households involved in the other monitoring activities [32]. As households had to agree to feed and maintain the cats, it was not possible to increase this number; (III) anticoagulant rodenticides (Lanirat containing bromadiolone, Novartis Ltd., Dhaka, Bangladesh), three bait stations per household with weekly bait replacement; and (IV) daily rodent trapping with four traps per household (using two snap traps of 14 × 7 cm; Big Snap-E (Kness, Albia, IA, USA), and two locally made live single capture cage traps measuring 10 × 15 × 33 cm). All traps were baited with banana. Traps were placed in the afternoon (between 15:00–17:00 h) and re-visited the next morning (between 09:00 and 11:00) to check for captures. Rodent species were identified according to Aplin et al. [33,34]. Each session took three months (June–August, and October–December), in which four villages were visited, receiving one of the four management methods. In each month, rice losses were assessed and rodent presence was monitored. In 2016, monitoring consisted of measuring GUDs, rice basket losses and tracking tiles; however, in 2017, a third method of trapping every fortnight for one night in all involved households was added to monitor rodent species’ presence. In month two and three the rodent control treatments were conducted alongside the monitoring activities.

### 2.5. Rodent Control Measures in Rice Mills

Two rice mills were selected for the study. At mill one, rodent control was conducted using domestic cats. At mill two, no rodent-management was applied (control). Starting in July 2016, two months of baseline monitoring losses and rodent presence data were collected. Starting in September, 20 cats were placed at rice milling station no. 1. The number of cats was agreed upon with the mill owner who would have to feed the cats. However, the number seemed reasonable based on the size of the mill and from what is currently known about domestic cat foraging and home range size [35,36,37]. Thereafter, the rice loss and rodent monitoring was continued for five more months (both locations, September 2016–January 2017). During months 8 and 9 (February and March 2017) in both mills, rodent presence was measured by rodent trapping for two consecutive days each fortnight (20 traps in total consisting of 10 snap and 10 live traps (Big Snap-E, Kness), and locally made live single capture cage traps measuring 10 × 15 × 33 cm). Rodent trapping was conducted in the warehouse of each milling station where paddy rice is stored in jute bags.

### 2.6. Data Analyses

The efficacy of rodent management (presence of cats) in rice mills was assessed using descriptive statistics only due to lack of replication of the cat treatment, i.e., only one mill with cats and one mill without cats was followed over time; thus the number of trapped rodents was shown simply split by rodent species and treatment.

Rodent management in households was empirically replicated, permitting more detailed analysis. We first present frequency tables of number of captured rodents split by season, village, year, and rodent species, and tables of average rice loss split by village, year, and time interval. The effect of rodent management method was studied using a modelling approach for three parameters: (i) amount of rice eaten by rodents per day, (ii) number of cells (out of 16) per tile with rodent footprints per night, and (iii) GUD (number of peanuts (out of 25) remaining per night). To quantify the strength of the pairwise relationships, Spearman rank correlation coefficients were calculated among rice loss measurements, percentage of tiles marked with footprints (averaged over locations within household and two consecutive days), and GUDs (averaged over two consecutive days).

All three parameters comprised repeated measurements per household, calling for appropriate statistical methods that handle correlated responses. To this end we used (generalised) linear mixed models ((g)lmm), which are described further below. To emphasise the repeated measurements per household, we produced scatterplots per year-season combination, showing data with observations from the same household connected by lines and coloured by treatment.

The repeated measurements on the three parameters were obtained as follows: (i) Per household using weight of baskets filled with rice, recording weights every two weeks. Weights were transformed into the average amount of rice eaten per day (over the past interval), and log-transformed as y = log (amount per day +1), leading to approximate normality and constant variance in later analysis; (ii) Per household using two tiles in two locations scored on the two mornings after the rice basket weight measurement; (iii) Per household using the GUD data that were recorded twice on the same two mornings.

The rice-filled baskets were monitored in all households without being assigned a treatment for four weeks (season 2016-wet) or after two weeks (other seasons), and then assigned a specific treatment at the village level (control, cats, rodenticide, traps). We labelled the time of application of the treatment as t = 0. After t = 0 five (2016-wet), two (2016-dry), or four (2017-wet, 2017-dry) data collection sessions were taken. Longer monitoring was not possible as most farmers do not normally store rice longer than three months.

The year-season combination was studied in four villages with all ten households per village receiving the same treatment because rodent control treatments had to be delivered at the village level. Therefore, the effects of village and treatment may be confounded. Potential inherent differences between villages were corrected through the analysis of baseline measurements taken before treatments were implemented to estimate the variability of the responses between villages.

We used an lmm for the log-amount of rice eaten per day, assuming a normal distribution, and glmms for the counts of marked cells and for the GUD, assuming binomial distributions with logit link function. The fixed and random parts for these three (g)lmms were largely identical and are described below.

The fixed part of the (g)lmms comprised treatment specific quadratic time trends, which were allowed to be different for year-season combinations. Per year-season combination, these time trends started from a common intercept at t = 0, as this was the time point when the treatment started. For the starting phase, a common quadratic time trend was assumed per year-season combination. Checks were made to see whether cubic time trends were needed.

In the random part of the (g)lmms we introduced the following components: (1) random quadratic time trends per household to handle the repeated measurements (which makes this mixed model a random coefficient model), (2) random effects for village-year combinations (largely allowing for differences in rodent population sizes between villages per year), (3) random effects for village-year-time combinations (allowing for deviations from a quadratic time trend at village level).

In the binomial glmms extra random effects were introduced to handle binomial overdispersion.

After fitting the (g)lmms we answered the following questions:

(a)Are the treatment time trends different in the four year-season combinations? If so, which year-season combinations are different?(b)Are the time trends different between treatments within year-season combinations? If so, which treatments have different time trends?(c)Are treatment effects different at specific timepoints within year-season combinations? We checked here at the timepoints when rice measurements were taken. If so, which treatments are different?

To answer these questions for the lmm, hypothesis tests were made using approximate F-tests with the Kenward-Roger method (a), followed by user-defined contrasts using F-tests (b), followed by post-hoc comparisons with the Tukey method (c). For the glmms, likelihood ratio tests were done (a), followed by user-defined comparisons using chi-square tests (b), and pairwise comparisons using Wald tests with the Tukey method (c). Results for these approaches are available as supplemental material.

All statistical analyses were performed using R version 3.6.1 (R Core team, Vienna, Austria); (g)lmm were fitted using package lme4 [38] and compared using package pbkrtest [39]; user-defined contrasts were made using package car [40]; treatment comparisons were made using package emmeans [41]; plots were produced using package ggplot2 [42].

## 3. Results

In total, 210 rodents were captured from inside people’s homes (Table 1). *Rattus rattus* was present in almost all villages, and trapped most often (43.3%), followed by *Mus musculus* (35.7%) and *Bandicota bengalensis* (12.4%).

In 2016 villages A-D were visited nine times, whereas villages E-H were visited five times. For 2017 adjustments to the original planning were made such that all villages (A–H) were visited six times. In 2016, all villages experienced similar losses, ranging from 677.9 g loss per basket per month to 846.5 g loss per month (13.6–16.9%) from the basket stored within the household (Table 2). In 2017, the losses were lower, ranging from 29.1 g per month to 107.9 g eaten per month (0.6–2.2%).

When comparing the three monitoring methods against the four treatments, the rice loss and tracking tiles showed the highest Spearman rank correlation (r = 0.75, *n* = 959), followed by GUDs and tracking tiles (r = 0.73, *n* = 1120), and rice loss and GUDs (r = 0.63, *n* = 959, see also in Appendix A). Treatments that are significantly different at individual time points (days from start of intervention) indicate which monitoring methods are generally more effective in assessing the impact of the rodent management treatments (Table 3). As expected, treatments are more likely to be different from each other at later time points.

### 3.1. Stored Rice Losses

#### 3.1.1. Villages

The treatment trends of daily rice loss over time showed significant differences among year-season combinations (Figure 5), when comparing all four year-season combinations simultaneously (*p* < 0.0001, F = 83.9, df = 27 and 9.7) as well as individual pairwise comparisons (all six pairwise comparisons *p* < 0.0001, see Appendix A, p. 6–8).

An analysis of data from the wet season of 2016 (Figure 6) indicated that rodent management interventions were different (*p* = 0.0017, F = 9.7, df = 6 and 8.9, see Appendix A, p. 9).

When comparing the responses on rice loss between the treatments at specific time points, significant differences were found. At 14 days after the start of the treatments, the untreated control group differed from the three treatment types and showed unexpectedly significant lower loss of rice per day (Figure 6). Over time the difference reversed, with lower loss of rice per day in the treatment groups. At 67 days after the start of the treatment (the final measurement day), significantly more rice was eaten per day in the control group than in the rodenticide group (t = 4.45, *p* = 0.006, df = 29.7) and trap treatment group (t = 4.79, *p* = 0.0002, df = 29.7), while the difference between the control and cat group was not significant (t = 2.13, *p* = 0.1658, df = 29.7, see Appendix A, p. 12). By the end there were also differences between the three treatments; the use of traps resulted in less rice loss per day compared to the use of cats (t = 2.695, *p* = 0.046, df = 49.3).

#### 3.1.2. Tracking Tiles

The treatment time trends assessing the percentage of tiles marked with rodent footprints on tracking tiles placed in the villages differed significantly amongst year-season combinations when comparing all four year-season combinations simultaneously (*p* < 0.0001, Χ^2^ = 323.4, df = 27) as well as through pairwise comparisons (Figure 7; *p* < 0.0001 for all pairs, see details in Appendix A, p. 22–25).

For the tracking tile data collected, there were differences in rodent activity observed over time among treatments in both wet seasons (2016 *p* < 0.0001, Χ^2^ = 29.5 df = 6, and 2017 *p* < 0.0001, Χ^2^ = 33.0, df = 6). When comparing the percentage of tiles marked with footprints, the final assessments at 56 and at 67 days since the start of the trial in the wet season of 2016 showed that there was significantly more rodent activity in the untreated control treatment than in the other treatment villages (Figure 8).

Tracking tile data from the wet season of 2017 indicated there were significant differences in treatment effects on the tracking tiles for 14, 28, and 42 days after the start of the treatments (see Appendix A, p. 30–38). The most effective method for 14, 28, and 42 days after the start of the treatments was the use of rodenticide.

#### 3.1.3. Giving Up Densities

The treatment time trends assessing the percentage of tiles marked with rodent footprints on tracking tiles placed in the villages differed significantly amongst year-season combinations when comparing all four year-season combinations simultaneously (*p* < 0.0001, Χ^2^ = 224.3, df = 27).

Data on the number of peanuts eaten by rodents from the GUD monitoring in village households indicated the treatments varied over time during the two seasons of 2016 (*p* < 0.0001, Χ^2^ = 115.78, df = 9 Figure 9), but that there was no such observed difference in the 2017 wet and dry seasons *(p =* 0.114, chi 14.23, df = 9, more details can be found in the Appendix A, p. 46).

When comparing the GUD data at specific time points, significant differences were found only in the wet season of 2016 at the final data collection times of 56 and 67 days after the start of the trial, where the untreated control group showed significantly higher GUDs than the trap treatments (at 56 days: *p* = 0.002, Z = 3.6, at 67 days: *p* = 0.0036, Z = 3.4) (Figure 10).

#### 3.1.4. Rice Milling Stations

In the rice mills, there was no substantial effect observed from the introduction of cats (Figure 11). Although it was not possible to replicate the data with different sites or seasons, the trend in rice losses did seem to reduce by 5–10% shortly after the placement of the cats in the mill. Furthermore, mill workers made the observation that it seemed there were fewer rodents around after the cats were introduced.

The tracking tiles and the GUDs placed in the mills showed no clear patterns and no changes after introducing the cats in one of the two mills. However, during the trapping phase in the end of the study, more rodents were trapped in the untreated control mill (48 rodents) than in the mill where cats were introduced for rodent management (20 rodents), with the same trapping intensity at both sites (Table 4).

## 4. Discussion

All households across all villages experienced similar losses, ranging from 13.6% to 16.9% loss per month from the baskets stored within the household in 2016, and in 2017 the losses were lower, ranging from 0.6% to 2.2% per month. Discussions with farmers suggest the lower losses experienced in 2017 may be due to more severe monsoonal flooding between these years in comparison to the monsoon prior to 2016, which resulted in less flooding. Flood waters will naturally reduce rodent habitat and rodent numbers, and thus may account for the differences observed in our data between the two years [43]. National statistics indicate that in 2013 the annual mean consumption of rice per person per month was 14.3 kg, or approximately 500 g per day [44]. From our 2016 data showing an average loss of 796.6 g/month, the lost rice could feed one person for almost two days extra each month. Research of Htwe et al. [9] in Myanmar also found differences in loss of grain across seasons; in 2013 they observed losses of 14% and 8.2% in two different villages and a year later in 2014 they observed losses of 4% and 1.2% in the same communities. Seasonal variations in such measurements should be expected and can be attributed to a number of factors that would influence the amount of rice produced and rodent population abundance, e.g., severity of monsoon flooding or rice crop growing conditions and yield. Longer-term studies would be required to understand the drivers of inter-annual variations in post-harvest losses caused by rodents. A study in Laos on rice loss by rodents found that losses were higher in the dry season compared to the wet season (10.3% and 7.4%, respectively) [10]. Our data are inconclusive in this respect, with no clear trend in losses observed between wet and dry seasons, and this may be due to relatively stable year-round rodent populations found in Bangladesh villages compared to more fluctuating populations in Laos [8,45].

With respect to rodent management methods we found that the main rodent pests in village households were *Rattus rattus*, followed by *Mus musculus,* and *Bandicota bengalensis*. This is in line with findings from Bangladesh [46], India [47], Pakistan [48], and Myanmar [9,18], where *R. rattus* and *B. bengalensis* were also found to be the main rodent pests. During the dry season, fewer rodents were trapped (*n* = 89) than during the wet season (*n* = 121), which agrees with what is known about these rodent species’ breeding and population dynamics [34].

It was expected that there would be steeper declines in the amount of rice in the baskets in the untreated control households compared to locations where a rodent management treatment was conducted. However, this was only observed in the wet season of 2016, where time trend differences were found between treatments. In order to make solid statements about which rodent management method is most efficient, the monitoring period should have been at least one month longer, as we began to see an effect near the end of the monitoring period (67 days after starting the treatments). However, many farmers do not store their rice longer than this, so extending the monitoring may lead to further confounding evidence. Our data on the time taken to observe potential benefits of rodent control on post-harvest losses suggest that management impacts may be more strongly observed if rodent control was implemented several months before rice was harvested and stored, allowing time for the rodent population to reduce before the storage period. From the management options applied, the use of rodenticides and daily trapping resulted in lower losses of rice in the baskets. As no statistical difference between these two management methods was observed, we recommend the use of daily trapping as a rodent pest management tool, as this is a non-toxic, sustainable method that costs less than rodenticide use, particularly as purchased traps can last for many years. Furthermore, the use of rodenticides is known to have a negative impact on the environment, where non-target species, including humans, can be affected [49,50], and where problems with development of rodenticide resistance are growing globally [51]. Other studies on rodent management methods also found daily rodent removal trapping to be effective [8,52,53]. A study from Uganda showed that the impact of trapping was highly effective but that the benefits disappear shortly after cessation of trapping [52]. Therefore, it is recommended that farming communities establish continuous trapping programmes in order to keep the rodent pest population at low levels. The intensity of trapping in terms of number of traps per household and type of trap to use would require some further investigation in collaboration with farming communities.

For both the villages and the rice mill, there was no significant effect of the placement of cats on the amount of rice lost from the baskets. However, when looking at the data from the mill, we observed that cat presence had a slight effect on the basket weight loss shortly after the introduction of the cats. Although the cats were fed daily to keep them in and around the households and mill, we think many of the cats strayed away during the trial monitoring period, and both the feeding and the dispersal of cats are likely to have influenced their potential predation impact. New innovations in low energy blue tooth tracking systems and GPS collars could be used to increase our understanding of domestic cat foraging activity and home ranges [49,50]. Despite the fact that some rodent species in Bangladesh are larger than cats, e.g., *Bandicota indica*, the presence of domestic predators could make effective contributions to an integrated rodent management strategy [51,52]. However, more research is required to understand the potential impacts of domestic cats on rodent pests in different local contexts. Mahlaba et al. [51] found no effect on rodent feeding in a GUD study when cats alone were present around homesteads; however, when cat presence was combined with the presence of dogs around homesteads, GUD data showed significant reductions in rodent foraging activity.

In our study the baskets with rice were not topped up after each measurement period, and it could be argued that this could reduce the attractiveness of rodents to the baskets over time through increased contamination with rodent droppings and urine. However, the farmer granaries where the baskets are placed are also subject to increased contamination over time, so topping up the baskets every fortnight could change the relative attractiveness of the baskets compared to the rice present in the farmers’ own granary, thus increasing the attractiveness of the rice in the baskets over time. Although this aspect of the methodology requires further research using context specific choice tests, we argue that the over-riding feeding behaviour will be influenced more strongly by relative contamination levels between rice in the basket and the farmer granary, which is more adequately controlled by not topping up the baskets. Another parameter with which it is difficult to control potential variation is biotic/abiotic conditions between the villages. All the villages involved were less than 10 km apart with similar socio-economic, cultural, geographic, and agricultural conditions. However, involved households varied from each other with respect to distance to roads, ponds, cropping areas, abundance of trees, etc., that could potentially skew variations in rodent abundance. We argue that our analysis has been able to correct for this in the glmm model due to our collection of monitoring data for 2–4 weeks before rodent management interventions started in each season and year.

When comparing the three rodent monitoring methods we found that the GUDs and tracking tiles showed similar results. Based on our results we would suggest to use tracking tiles for monitoring as they are relatively easy to implement, whilst other research confirms that rodent activity measured with tracking tiles correlates very well with rodent abundance [29]. Another benefit of using tracking tiles is that rodent activity is passively measured, compared to the use of GUDs or rice baskets, which may attract rodents, or removal trapping that will reduce the population. The relative weakness of the GUDs in comparison to the other methods used may be due to the fact that they are in feeding competition with the much larger farmer grain store where the rice baskets are and where the rice is provided entirely open without the need to spend effort finding the peanuts in the GUD. However, GUDs can provide insights into the feeding behaviour and habitat preferences of animals by giving an index of the costs of foraging in a given area as well as changes in population abundance [53]. Unfortunately, our use of different monitoring methods was unable to untangle the potential effects of predation by cats on reduced foraging by rodents vs. lower rodent populations induced by trapping or rodenticides. Besides predation, the presence of natural predators can induce innate fear responses [54,55]. Predator cues such can be through vision, olfaction, and sensing [56]. In the current study, there could have been an effect of the cat cues on the rodent presence, although it would have been difficult to truly test these effects due to the low replication of the cat sites. For further research it would be valuable to measure the effect of predator presence on rodent presence.

In conclusion, our data confirm that post-harvest losses caused by rodents in rural communities are significant and exacerbate food insecurity and food safety issues. We recommend that rodent pests are continuously controlled by the use of snap traps, which will ensure the rodent population is already low at the start of each storage period. Improving household granaries in Bangladesh to make the structures rodent proof through the addition of lids could also reduce stored food losses and increase human health.

## Figures and Tables

**Figure 1 animals-10-01612-f001:**
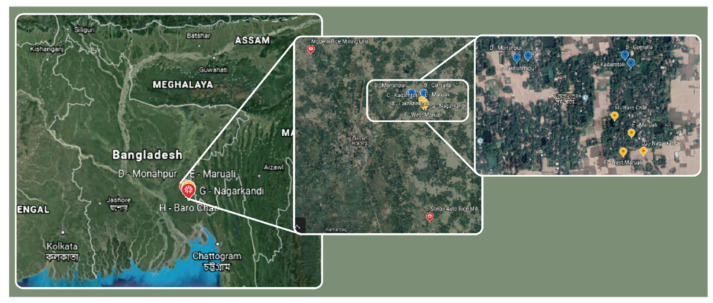
Map showing locations of the participating mills and the villages in Bangladesh, based on GPS coordinates. Villages A–D are marked yellow, E-H blue, and the mills are marked red.

**Figure 2 animals-10-01612-f002:**
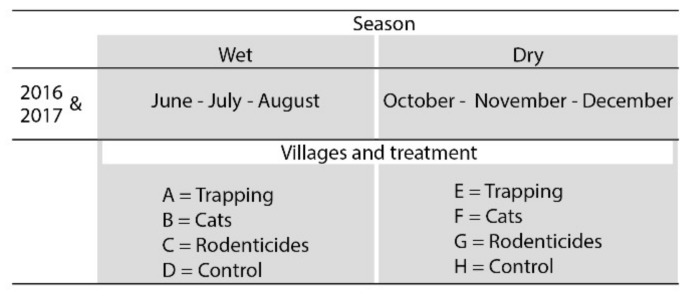
Overview of the study design of eight selected villages in Bangladesh.

**Figure 3 animals-10-01612-f003:**
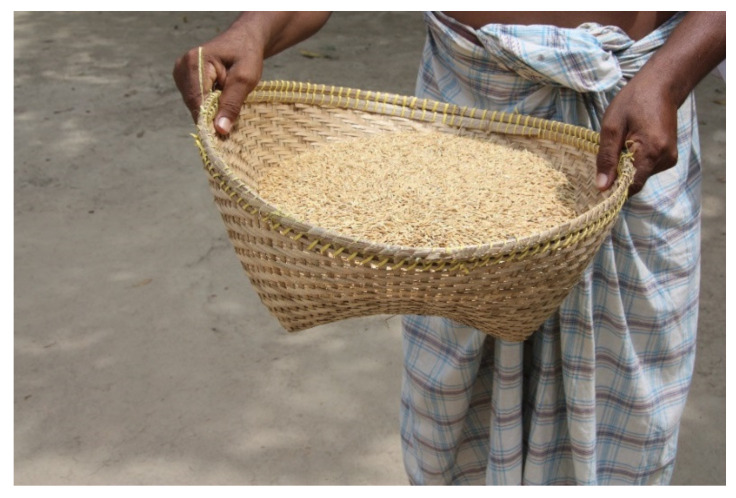
Baskets made from woven reeds and bamboo filled with rice to assess the rice losses due to rodents.

**Figure 4 animals-10-01612-f004:**
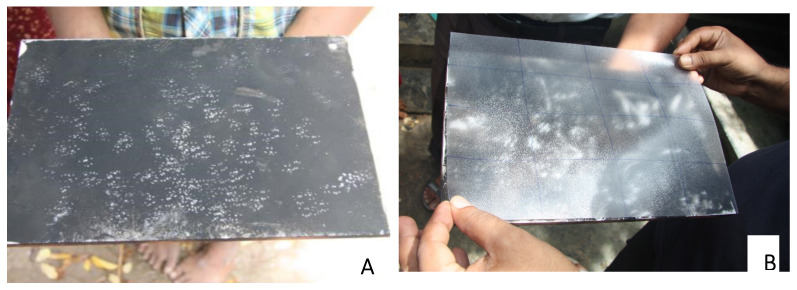
Tracking tiles: (**A**) with rodent footprints; (**B**) determining the percentage area marked by rodent footprints by placing a transparent plastic sheet marked into 16 cells on top of the tile.

**Figure 5 animals-10-01612-f005:**
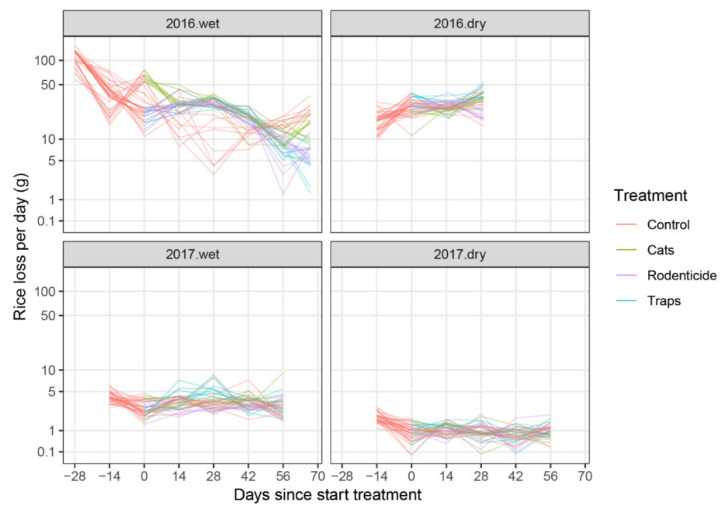
Figure **5.** Amount of rice loss per day by rodents taken from baskets placed in farmer stores, split by year and season. Lines connect observations from the same experimental units (baskets) over time (days).

**Figure 6 animals-10-01612-f006:**
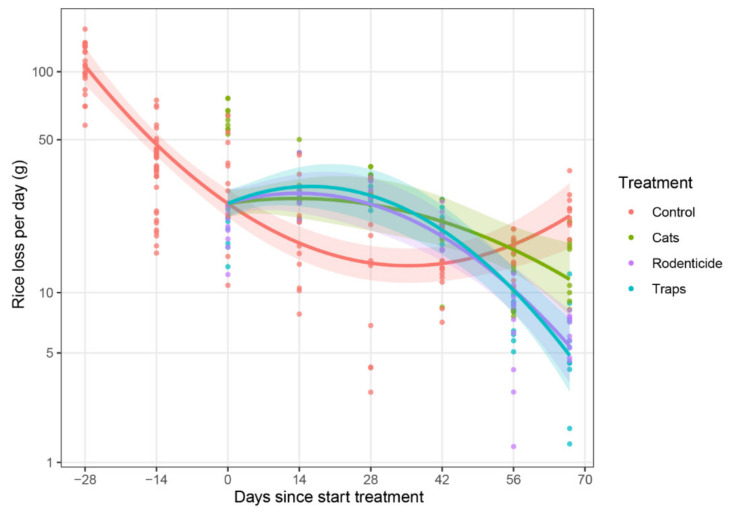
Rice loss by rodents per day from baskets of rice in farmer stores during the wet season of 2016. Estimated time trend lines for each treatment were generated by a mixed model (see Section 2.6), with shaded areas representing 95% confidence bands, indicating that treatments had an effect on reducing rodent losses compared to the untreated control by the end of the storage period.

**Figure 7 animals-10-01612-f007:**
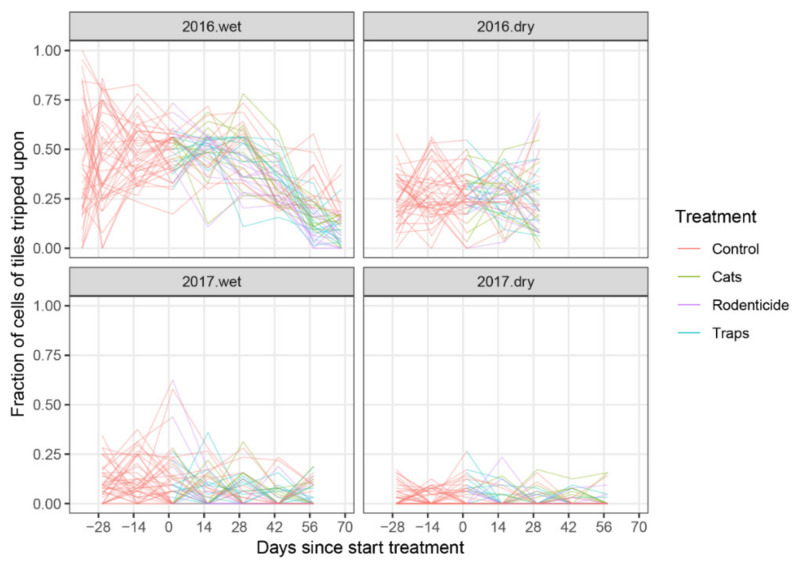
Percentage of tracking tiles marked with rodent footprints, using average values from two locations and two consecutive observation days (*n* = 4), split by year and season. Lines connect observations of rodent activity from the same households over time (days).

**Figure 8 animals-10-01612-f008:**
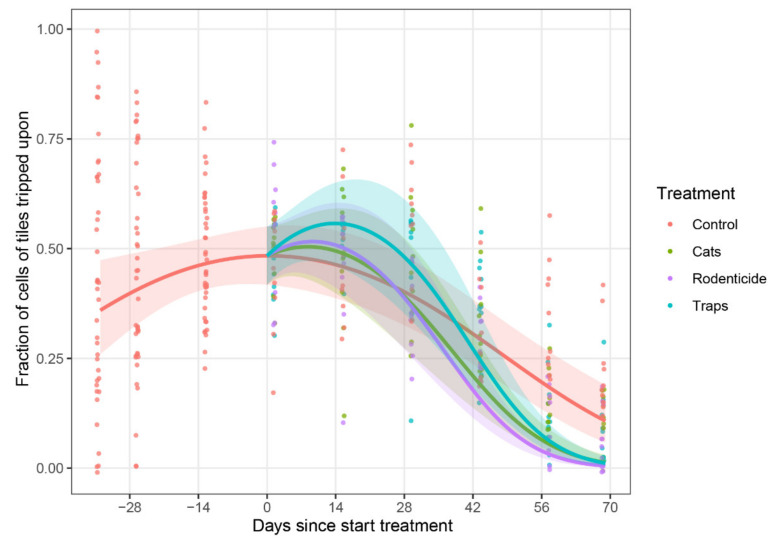
Percentage of tracking tiles marked with rodent footprints, using average values from two locations and two consecutive observation days (*n* = 4), in the wet season of 2016. Estimated time trend lines for each treatment were generated by a generalised linear mixed model (described in M&M), with shaded areas representing 95% confidence bands for the predicted mean response, indicating that treatments had an effect on reducing rodent losses compared to the untreated control by the end of the storage period.

**Figure 9 animals-10-01612-f009:**
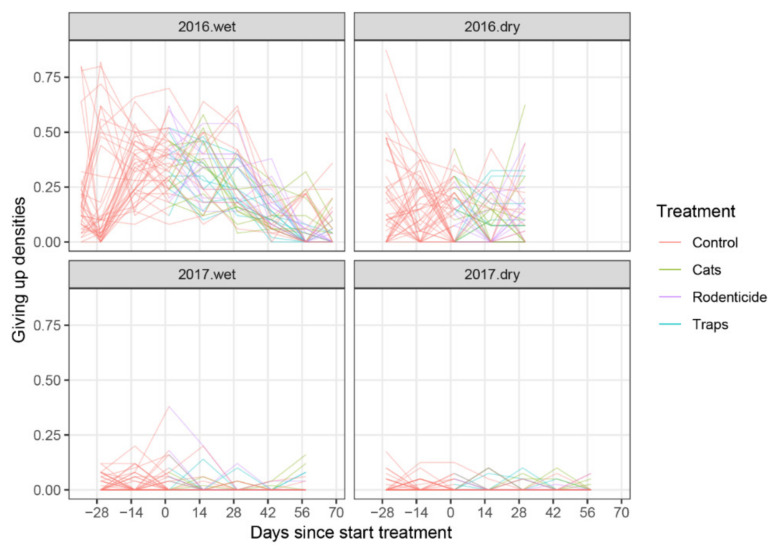
Percentage of peanuts eaten by rodents per day (averaged over two consecutive observation days) in a giving up density monitoring trial, split by year and season. Lines connect observations from the same household over time (days).

**Figure 10 animals-10-01612-f010:**
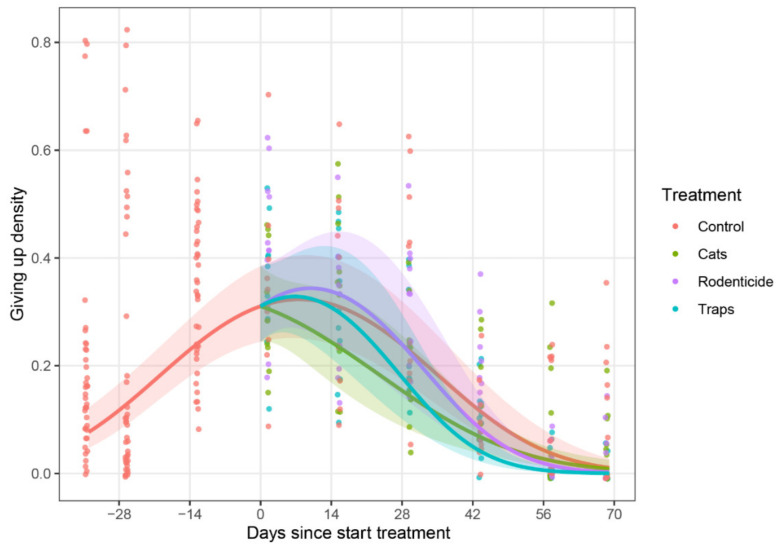
Percentage of peanuts eaten by rodents per day (averaged over two consecutive observation days) in a giving up density monitoring trial in the wet season of 2016. Estimated time trend lines for each treatment were generated by a generalised linear mixed model (described elsewhere) with shaded areas representing 95% confidence bands for the predicted mean response. By the end of the storage period, the number of peanuts eaten from the GUD in the trapping and rodenticide treatments was marginally lower than the number eaten in the untreated control GUD.

**Figure 11 animals-10-01612-f011:**
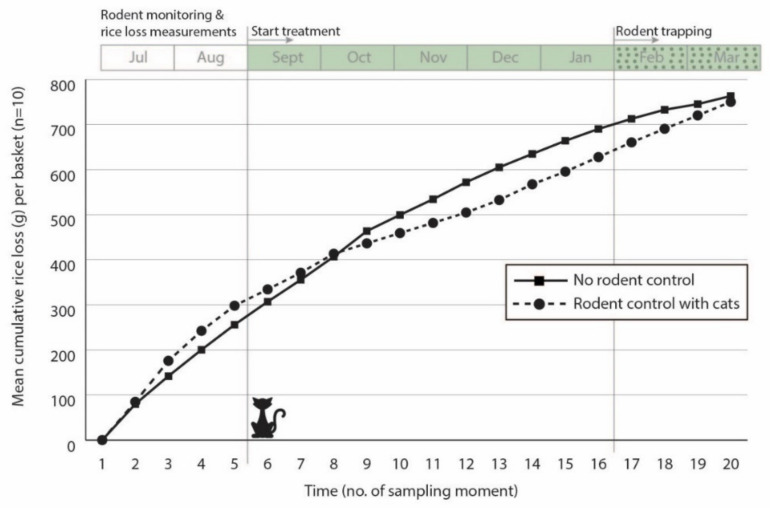
Fortnightly measurements of the cumulative weight loss (g) of stored rice removed by rodents from baskets placed in two rice mills (one without rodent control and one with rodent control by cats) from July 2016 to March 2017.

**Table 1 animals-10-01612-t001:** Number and species of rodents captured inside households of the participating villages in Bangladesh, over the wet and dry seasons of 2016 and 2017. The village letters A to H are between brackets after the village names.

Season.	Village, Year	Treatment	Total Number Captured	*Bandicota bengalensis*	*Bandicota indica*	*Mus musculus*	*Mus terricolor*	*Rattus exulans*	*Rattus rattus*
Wet	Laksmipur (A)	Trapping							
	2016		64	10	0	24	0	0	30
	2017		21	0	0	13	0	5	3
	Comalla (B)	Cats							
	2017		8	1	0	2		0	5
	Kadamtoly (C)	Rodenticides							
	2017		5	0	0	0	1	0	4
	Monahpur (D)	Control							
	2017		23	11	0	3	0	0	9
		Subtotal	121	22	0	42	1	5	51
Dry	Maruali (E)	Trapping							
	2016		42	0	6	20	0	2	14
	2017		39	3	3	10	1		22
	West Maruali (F)	Cats							
	2017		3	0	0	2	0	0	1
	Nagar Kandi (G)	Rodenticides							
	2017		2	1	0	1	0	0	0
	Baro Char (H)	Control							
	2017		3	0	0	0	0	0	3
		Subtotal	89	4	9	33	1	2	40
		Total	210	26	9	75	2	7	91

**Table 2 animals-10-01612-t002:** Average amount of stored rice-loss in Bangladesh households per interval (14 days), *n* = 10 baskets per village.

	Mean (± SD) Losses (g) Per Village (Treatment Between Brackets)							
	2016							
Interval	Laksmipur (Trapping)	Comalla (Cats)	Kadamtoly (Rodenticides)	Monahpur (Control)	Maruali (Trapping)	West Maruali (Cats)	Nagar Kandi (Rodenticides)	Baro Char (Control)
1	1046.8 ± 13.4	690.0 ± 133.3	1051.4 ± 11.7	963.9 ± 140.7	224.2 ± 83.1	255.5 ± 81.0	227.2 ± 45.7	270.4 ± 52.6
2	557.9 ± 58.1	294.0 ± 61.1	554.1 ± 53.9	781.2 ± 175.9	459.9 ± 90.0	330.9 ± 83.3	374.9 ± 56.0	367.2 ± 66.2
3	290.9 ± 57.6	889.1 ± 117.1	291.5 ± 62.3	495.7 ± 237.7	408.4 ± 75.4	349.1 ± 35.2	434.4 ± 76.2	398.3 ± 32.1
4	400.9 ± 85.9	433.3 ± 127.1	398.0 ± 86.8	274.4 ± 159.0	572.5 ± 91.5	465.7 ± 72.1	319.2 ±71.3	451.9 ± 148.0
5	417.1 ± 57.8	451.7 ± 62.4	416.5 ± 35.6	206.5 ± 149.4				
6	299.7 ± 40.3	262.1 ± 68.1	270.1 ± 51.5	173.6 ± 48.4				
7	111.9 ± 30.9	159.8 ± 48.0	101.3 ± 49.3	219.5 ± 45.4				
8	61.3 ± 35.7	162.8 ± 53.9	71.0 ± 14.5	270.9 ± 56.6				
	**2017**							
	**Laksmipur**	**Comalla**	**Kadamtoly**	**Monahpur**	**Maruali**	**West Maruali**	**Nagar Kandi**	**Baro Char**
1	60.5 ± 11.5	60.1 ± 12.1	54.0 ± 8.1	61.7 ± 13.3	32.4 ± 5.4	24.5 ± 2.4	22.5 ± 3.0	27.8 ± 5.2
2	36.8 ± 9.3	42.6 ± 15.1	38.4 ± 9.4	38.7 ± 12.8	15.2 ± 5.9	14.7 ± 6.5	11.8 ± 5.8	11.2 ± 6.1
3	60.2 ± 19.4	47.1 ± 9.3	41.7 ± 11.3	50.9 ± 10.0	17.9 ± 4.1	15.4 ± 4.8	13.4 ± 5.4	13.9 ± 4.9
4	78.6 ± 22.0	45.5 ± 8.9	40.9 ± 7.1	51.8 ± 22.3	14.3 ± 6.0	12.6 ± 6.8	14.1 ± 8.0	14.6 ± 4.3
5	44.7 ± 11.1	51.7 ± 9.8	42.9 ± 6.6	53.0 ± 22.2	11.6 ± 5.7	9.5 ± 3.4	10.1 ± 7.2	11.6 ± 4.7
6	43.1 ± 17.5	51.3 ± 28.5	44.2 ± 12.1	32.7 ± 9.2	14.8 ± 5.4	11.2 ± 4.5	16.6 ± 5.3	13.4 ± 5.6

**Table 3 animals-10-01612-t003:** Time points (t = days from start of intervention) where comparisons across treatments indicated that some monitoring methods are more likely to point to statistical differences among treatments than others. Pairwise comparisons between the four treatments were made for each timepoint, using the Tukey method with α = 0.05. Values in the same column followed by the same letter are not statistically different from each other.

		2016-Wet			2017-Wet
		Rice Loss	Tiles	GUD	Tiles
t = 14	Control	a	a	a	c
	Cats	b	a	a	a,b
	Rodenticide	b	a	a	a
	Traps	b	a	a	b,c
t = 28	Control	a	a	a	c
	Cats	a,b	a	a	a,b
	Rodenticide	a,b	a	a	a
	Traps	b	a	a	b,c
t = 42	Control	a	a	a	b
	Cats	a	a	a	a,b
	Rodenticide	a	a	a	a
	Traps	a	a	a	b
t = 56	Control	a	b	b	a
	Cats	a	a	b	a
	Rodenticide	a	a	a,b	a
	Traps	a	a	a	a
t = 67	Control	c	b	b	
	Cats	b,c	a	b	
	Rodenticide	a,b	a	a,b	
	Traps	a	a	a	

**Table 4 animals-10-01612-t004:** Rodents trapped in two rice mills in Bangladesh.

	Rice Mills	
Species	Control	Cats
*Rattus rattus*	12	7
*Mus musculus*	16	5
*Bandicota bengalensis*	13	4
*Rattus exulans*	7	1
*Mus terricolor*	0	3
Total	48	20

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
