# Peer review of "Efficacy of Management and Monitoring Methods to Prevent Post-Harvest Losses Caused by Rodents"

_animals, 2020, doi:10.3390/ani10091612_

Round 1

Reviewer 1 Report

The manuscript “Efficacy of management and monitoring methods to 2 prevent post-harvest losses caused by rodents“ addresses the post-harvest rice losses and efficacy of counteractions. The authors conducted extensive data acquisition by a sophisticated study design and evaluated data by sound statistical methods.

In my opinion, the Materials and Methods section provides comprehensive information on the well thought out study design. Results are supported by the figures. The discussion addresses all relevant issues of the results and compares the results critically to outcomes from other publication.
All parts of the manuscript are well written and I think there are only very few slips which need to be corrected.

Some minor remarks:
- Simple summary and abstract: It is not clear why the losses in 2016 and 2017 differ distinctly. Was 2016 first year of application of the control strategies or a "control year" without any measures like trapping, rodenticides etc.? I think it would be good to state in the summary already that the latter was not the case by giving the years when you mention the “two year period”.
- L 31/32 + L 46 Mus musculus in italic
- L 35 + L 50...removal BY trapping....
- L 69, 74, 76: The way of citing is not homogenous.
- L 107: E) Maurali or Maruaki?
- L 129: add ~80% "of households"
- Figs. 5 - 10: You could adjust the scaling of the x-axis to the time points of measurement i.e. every 14 days. Then it would be easier to remember from methods section that it was not continuous measurement. In Fig. 11 the measuring period becomes clearer.
- Unfortunately, I had no access to the supplementary material. I could not check and it became not really clear from the text if sufficient diagnostic plots or other data for the models were included. If not, it would be great to add these.

Reviewer 2 Report

This is an important paper, concerning a topic of world pest management and food security interest. The authors studied the different effects of four rodent control strategies: control, domestics cats, rodenticides and snap-traps on rodent abundance and stored produce loss in Bangladesh. The most effective method of reducing produce loss was snap-trapping, although rodenticides and cats appear to have some effects on rodent abundance.

The manuscript was well-written, and the study was well-designed considering the logistical limitations. Photographs do help the reader understand the study and the figures are clear. I only have minor comments to address for the manuscript.

The authors used generalised linear mixed effects models to analyses the data. It would make the analyses more interpretable if the authors also gave the coefficients and the test statistics, not just the p values. These can be in tables, or with the p-values in the text. P-values do not mean much by themselves.

Cats appear to have a slight effect on rodent abundance, and it was noted that it was impossible to untangle the effects of rodent reduction via predation compared to changes in behaviour. Hence some mention should be made of the quite extensive literature of cat odours on rodent behaviour. Replication of the cat sites was low, so it would have been difficult to truly test these effects. However, some of the literature would be useful to cite in this area are listed below [1-3]

  1. Bedoya-Pérez, M.A.; Smith, K.L.; Kevin, R.C.; Luo, J.L.; Crowther, M.S.; McGregor, I.S. Parameters that affect fear responses in rodents and how to use them for management. Frontiers in Ecology and Evolution 2019, 7.
  2. Apfelbach, R.; Blanchard, C.D.; Blanchard, R.J.; Hayes, R.A.; McGregor, I.S. The effects of predator odors in mammalian prey species: A review of field and laboratory studies. Neurosci. Biobehav. Rev. 2005, 29, 1123-1144.
  3. Parsons, M.H.; Apfelbach, R.; Banks, P.B.; Cameron, E.Z.; Dickman, C.R.; Frank, A.S.K.; Jones, M.E.; McGregor, I.S.; McLean, S.; Müller-Schwarze, D., et al. Biologically meaningful scents: A framework for understanding predator–prey research across disciplines. Biological Reviews 2018, 93, 98-114.
